# AKT2 Loss Impairs BRAF-Mutant Melanoma Metastasis

**DOI:** 10.3390/cancers15204958

**Published:** 2023-10-12

**Authors:** Siobhan K. McRee, Abraham L. Bayer, Jodie Pietruska, Philip N. Tsichlis, Philip W. Hinds

**Affiliations:** 1Program in Genetics, Graduate School of Biomedical Sciences, Tufts University, Boston, MA 02111, USA; skmcree@gmail.com; 2Department of Developmental, Molecular and Chemical Biology, Tufts University School of Medicine, Boston, MA 02111, USA; jodie.pietruska@gmail.com; 3Program in Immunology, Graduate School of Biomedical Sciences, Tufts University, Boston, MA 02111, USA; abraham.bayer@tufts.edu; 4Department of Immunology, Tufts University School of Medicine, Boston, MA 02111, USA; 5Department of Cancer Biology and Genetics, The Ohio State University, Columbus, OH 43210, USA; tsichlis.1@osu.edu

**Keywords:** melanoma, metastasis, cancer, AKT, signaling

## Abstract

**Simple Summary:**

Skin cancer, such as melanoma, is often treatable but becomes deadly when it expands to other places in the body, a process called metastasis. While many current drugs target melanoma tumor growth, no drugs yet exist to specifically prevent metastasis. Improving treatment outcomes, therefore, will require an understanding of the molecular mechanisms leading to metastasis. The AKT family of proteins are important regulators of cellular growth and signaling, which play differing roles in cancer. This work sought to study each of the AKT isoforms and their contributions to melanoma cell migration and metastasis. We found that AKT2 specifically regulates melanoma cell metastasis through effects on metabolism and melanoma cell properties, while AKT1 is involved in cellular proliferation and growth. This study suggests that specifically targeting AKT2 and AKT1 represents novel therapeutic strategies for different-stage melanoma patients.

**Abstract:**

Despite recent advances in treatment, melanoma remains the deadliest form of skin cancer due to its highly metastatic nature. Melanomas harboring oncogenic BRAF^V600E^ mutations combined with PTEN loss exhibit unrestrained PI3K/AKT signaling and increased invasiveness. However, the contribution of different AKT isoforms to melanoma initiation, progression, and metastasis has not been comprehensively explored, and questions remain about whether individual isoforms play distinct or redundant roles in each step. We investigate the contribution of individual AKT isoforms to melanoma initiation using a novel mouse model of AKT isoform-specific loss in a murine melanoma model, and we investigate tumor progression, maintenance, and metastasis among a panel of human metastatic melanoma cell lines using AKT isoform-specific knockdown studies. We elucidate that AKT2 is dispensable for primary tumor formation but promotes migration and invasion in vitro and metastatic seeding in vivo, whereas AKT1 is uniquely important for melanoma initiation and cell proliferation. We propose a mechanism whereby the inhibition of AKT2 impairs glycolysis and reduces an EMT-related gene expression signature in PTEN-null BRAF-mutant human melanoma cells to limit metastatic spread. Our data suggest that the elucidation of AKT2-specific functions in metastasis might inform therapeutic strategies to improve treatment options for melanoma patients.

## 1. Introduction

Metastasis is responsible for the vast majority of all cancer deaths, yet no therapies preferentially target the metastatic process. A key hallmark of cutaneous melanoma is its rapid and efficient metastasis, facilitated, in part, by the PI3K/AKT pathway [1]. PI3K/AKT signaling is hyperactivated in late-stage melanomas [2], often through the loss of its negative regulator, PTEN. PTEN inactivation leads to unrestrained activity regarding the PI3K effector kinase, AKT, which co-operates with pre-existing oncogenic BRAF mutations to promote metastasis [3]. Indeed, while the BRAF mutation V600E hyperactivates the MAPK signaling pathway and is a very common early mutation, it alone is insufficient for tumorigenesis, and subsequent genetic and/or epigenetic alterations are required [4,5]. Alterations in PTEN, including inactivating mutations, epigenetic silencing, protein destabilization, or genetic loss, occur in as many as 50% of all melanomas and correlate with advanced disease, including brain metastasis [6,7]. Accordingly, AKT is hyperactivated in melanoma brain metastases [8], and the inhibition of the PI3K pathway effectively reduces metastasis in murine melanoma models [9]. Despite this, pan-PI3K inhibitors have not yet demonstrated clinical efficacy [10,11]. While six PI3K inhibitors have been approved for clinical use against breast cancer, hematologic cancers, and other diseases, only one exhibits pan-PI3K inhibition, whereas others are isoform-specific. [12] A similar spectrum of clinical efficacy is likely from the AKT inhibitors currently being used in clinical trials, with isoform-specific inhibition better tolerated compared to pan-inhibition. This suggests that the additional isoform-specific PI3K/AKT inhibitors under development might be more therapeutically effective than pan-inhibition strategies. [13,14]. 

The AKT kinase family comprises three highly homologous yet functionally distinct isoforms (AKT1, AKT2, and AKT3) [15,16,17,18]. They differentially contribute to tumorigenesis in the breast, colon, and prostate [19,20], among other tissues, but isoform-specific inhibition has yet to be exploited as a therapeutic strategy. AKT isoforms achieve their specificity through differential activation, preferential substrate phosphorylation, and tissue distribution [19], but how these differential properties contribute to distinct cell transformation events is still being elucidated. While the isoform-specific effects of AKTs have been demonstrated in numerous other cancers [21,22,23] and in melanoma [24,25,26,27], many questions remain regarding the contribution of individual AKT isoforms to melanoma progression or metastasis, and few studies have interrogated the contribution of individual AKT isoforms to melanoma initiation. Large genomic datasets have provided some insights as to the relative frequency of AKT-isoform-specific alterations; the activation of mutations or gene amplification of AKT1 and AKT2 are more common in BRAF^V600E^ mutant and PTEN-null melanomas [28,29], whereas AKT3 amplification is more common in BRAF^WT^PTEN^WT^ tumors [26], although AKT3 has also demonstrated co-operativity with BRAF^V600E^ to induce murine melanoma development [30]. With regard to melanoma metastasis, multiple and differing roles for AKT isoforms have been described in both mouse and human models. For example, upon the loss of PTEN, the invasiveness of human melanoma cell lines was enhanced through the preferential phosphorylation of AKT2, whereas AKT3 phosphorylation reduced the invasive potential of PTEN-null melanoma cells [31]. Similarly, it has been shown that the PHLPP1 phosphatase suppressed melanoma metastasis through the dephosphorylation of AKT2 or AKT3 but not AKT1 in human melanoma cells [25]. In Braf^V600E^-driven murine melanomas lacking Cdkn2a, AKT1 activation was shown to enhance brain metastasis, whereas AKT2 and AKT3 had a less pronounced effect [27,32]. While the totality of data is suggestive of AKT isoforms playing differing roles in melanoma initiation, progression, and metastasis, further clarification through additional studies is needed, especially given the importance of metastasis to melanoma outcomes. 

Here, we employ inducible AKT-isoform-specific shRNAs and CRISPR/Cas9-mediated gene editing in human melanoma cell lines to outline the role of AKT2 in the metastatic cell seeding of oncogenic BRAF-driven PTEN-null melanoma through the enhancement of cell migration and invasiveness. As such, AKT2 depletion impairs and prevents metastasis, improving overall survival in mice. We further investigate AKT isoform phosphorylation in primary spontaneous murine melanomas and human metastatic melanoma cell lines, finding that AKT2 phosphorylation specifically increases in metastatic lesions. We further show that while AKT2 is dispensable for melanoma initiation, AKT1 contributes to BRAF-driven murine melanoma initiation and cell proliferation, which is in line with our previous results [33]. Mechanistically, we find AKT2 promotes metastasis and regulates glycolysis through a PDHK1-PDHE1α axis. This work supports the continued investigation of targeted AKT inhibition as an anti-melanoma therapy and presents a novel potential target for preventing metastatic spread.

## 2. Materials and Methods

### 2.1. Mouse Strains

All mice were maintained in a heat- and humidity-controlled AAALAC-accredited vivarium operating under a standard light-dark cycle. All protocols have been approved by the Institutional Animal Care and Use Committee (IACUC) at Tufts University School of Medicine, where the mice were housed and the experiments were conducted. BRAF^V600E^; Arf^−/−^ mice were bred in-house, whereas the AKT isoform knockout mice have been described previously [23]. NOD/SCID and C57Bl6/J mice were purchased from the Jackson Laboratory (Bar Harbor, ME, USA).

### 2.2. Xenografts

Male NOD/SCID mice (Jackson Laboratory, 6–10 weeks old) were injected subcutaneously with 2 × 10^6^ human melanoma cells, according to approved protocols. Once palpable (after roughly 3–4 weeks, and ~200 mm^3^), by using light touch compared to non-inoculated flank, the tumors were measured 3× weekly using calipers, and tumor volume was calculated using the formula [(π/6) × L × W^2^]. Tumors were allowed to grow until a limit of 1500 mm^3^ or 2 cm in any single direction or until mice became moribund. Doxycycline chow (200 mg/kg, Teklad, Chicago, IL, USA) was introduced when tumors were palpable when indicated.

### 2.3. Luciferase Imaging

Mice were anesthetized with isofluorane and injected intraperitoneally with 10 µL/g of body weight of Luciferin (Thermofisher, Waltham, MA, USA) 5 min prior to imaging. Imaging was performed using an IVIS SpectrumCT in vivo imaging system and analyzed using Living Image^®^ Software, version 4.7.

### 2.4. Metastasis Assays

Human (1 × 10^6^) or mouse melanoma cells (0.5 × 10^6^) were injected into a lateral tail vein of NOD/SCID or C57Bl6/J mice, respectively. Mice were maintained on regular or doxycycline chow (200 mg/kg, Teklad), imaged at 2 weeks post-injection and weekly thereafter. Mice were euthanized when moribund.

### 2.5. Tumor Cell Isolation and Tissue Preparation

Tumors were minced and digested with 3 mg/mL collagenase and 250 U/mL hyaluronidase for 2–4 h at 37 °C. The contaminating red blood cells were lysed with RBC Lysis Buffer (Sigma), and the organoids were triturated using an 18 G syringe needle, incubated in 0.05% Trypsin/0.53 mM EDTA, passed through a 40 μM cell strainer, and cultured in RPMI1640/10% FBS/1% penicillin/streptomycin/fungizone.

### 2.6. Cell Lines/Tissue Culture

SM1 cells were a generous gift from Antoni Ribas (UCLA, Los Angeles, CA, USA), and the TUMM cell lines were generated as described above. Human melanoma cell lines were generous gifts from Frank Haluska (Tufts Medical Center, Boston, MA, USA) and were routinely validated for melanocytic identity by the RNA or protein expression of MITF-M and pigment enzymes TYR and DCT and tested for mycoplasma contamination. Human cells were maintained in DMEM (Invitrogen, Cambridge, MA, USA) with 10% FBS (Atlanta Biologicals, Flowery Branch, GA, USA) and 1% penicillin/streptomycin (Invitrogen). The SM1 and TUMM murine cell lines were cultured in RPMI1640/10% FBS/1%penicillin/streptomycin/fungizone. Knockdown experiments utilized doxycycline (Sigma, St. Oouis, MO, USA) at concentrations of 0.5–1 µg/mL. Doxycycline-inducible shAKT plasmids were generous gifts from Drs. Alex Toker and Rebecca Chin (Beth Israel Deaconess Medical Center, Boston, MA, USA) [34]. A nontargeting hairpin scramble sequence (Sigma) was cloned into the Tet-pLKO-puro backbone, a gift from Dmitri Wiederschain (Addgene plasmid #21915). The 293T cells were transfected with shRNA constructs and packaging plasmids (psPAX2 and VSV-G) using PEI (polyethylenimine MW25,000, Polysciences, Warrington, PA, USA). Viral supernatants were collected at 48 h and 72 h post-transfection and mixed. Stably transduced cell lines were generated by infection overnight in the presence of 8 μg/mL polybrene, and 48 h after infection, they were selected with 1 μg/mL Puromycin (Gibco, Cambridge, MA, USA) for three days. Cells infected with pLENTI-Luciferase-expressing virus (generous gift of Charlotte Kuperwasser, Tufts University) were selected with neomycin (G418, 500 μg/mL, Gibco) for 2–3 weeks. CRISPR knockout WM1799 cells were previously generated and cultured, as described [33].

### 2.7. Immunoblot and Immunoprecipitation Analysis

Cells were lysed in RIPA buffer containing protease and phosphatase inhibitors (Roche, Basel Switzerland) and cleared by centrifugation. Protein concentration was determined by DC Protein Assay (BioRad, Hercules, CA, USA), and equivalent masses of protein were resolved using SDS-PAGE and transferred to 0.2 µm PVDF membranes (BioRad) for immunoblotting with indicated antibodies (see Appendix A). Membranes were incubated with horseradish peroxidase-conjugated secondary antibodies and visualized using enhanced chemiluminescence (Pierce, Waltham, MA, USA). For immunoprecipitation, samples were lysed in RIPA or CST lysis buffer (Cell Signaling Technologies, Danvers, MA, USA), and protein concentrations were normalized to 1 mg/mL and precleared with Protein A magnetic beads (Pierce). The antibody or equivalent control IgG was incubated overnight at 4 °C, then with Protein A beads for 60–90 min. Antibody/bead complexes were washed extensively in lysis buffer, eluted by boiling in Laemmli buffer, and resolved by immunoblotting, as described.

### 2.8. Wound Healing Assay

Cells were pre-incubated with doxycycline (DOX, 1 µg/mL) or DMSO for 24 h; then, a wound was made vertically across the growth area using a P1000 pipette tip. The cells were then washed with PBS, and the media was replaced with DOX or DMSO, as indicated. The cells were immediately imaged using an inverted scope at 4× magnification, aligned to a horizontally drawn guide perpendicular to the wound to ensure consistent imaging. After 16 h, the plates were imaged again using the same guide. Scratch distances were quantified using ImageJ and expressed as a percentage of wound closure relative to the DMSO-treated cells.

### 2.9. Migration/Invasion Assays

Cells were pre-incubated with doxycycline (DOX, 1 µg/mL) or DMSO for 24 h, and then seeded (25,000 cells) in the upper chamber of trans-wells (8 μm pores, Corning, Corning, NY, USA) in serum-free DMEM with DOX or DMSO, using 10% FBS as a chemoattractant and incubated overnight. Inserts were washed with PBS, scrubbed, fixed in ice-cold methanol, and stained with DAPI. Invasion assays utilized growth factor-reduced (GFR) Matrigel-coated trans-wells (Corning) and a 36 h incubation time.

### 2.10. Anchorage-Independent Growth (Soft Agar) Assay

Sterile low melting agarose (SeaPlaque, Lonza, Basel Switzerland) was prepared at a 5% stock concentration, with a 1% final concentration (bottom layer) prepared by dilution in an appropriate cell culture medium; this was allowed to solidify at room temperature for 30 min and was overlaid with 10,000 cells per 6 wells in 0.5% agarose. After solidifying for 30 min, the wells were overlaid with 0.5 mL of media containing 0.5 µg/mL of DMSO or doxycycline and allowed to incubate at 37 °C for 2–3 weeks or until macroscopic colonies were visible. The media was refreshed every 2–3 days. The plates were then fixed with 10% neutral buffered formalin, washed 1× with PBS, stained with 0.05% crystal violet overnight, and washed with PBS until clear. Images were taken at 10× magnification and quantified using ImageJ.

### 2.11. Seahorse Glycolytic Rate Assay

The WM1799 cells were pre-incubated with DMSO/DOX for 48 h, and 21,000 cells were seeded into microtiter plates (Agilent, Santa Clara, CA, USA) 1 day prior to the assay. One hour prior to the assay, the cells were washed and incubated with RPMI at pH 7.4 (Agilent) and placed in a non-CO_2_ incubator. The assays were performed using a Seahorse XFe96 Analyzer and Wave Software 2.4.0.

### 2.12. Quantitative RT-PCR

The cells were lysed in TRIzol (Invitrogen), and the RNA was isolated by phenol-chloroform extraction according to the manufacturer’s instructions, followed by lithium chloride/isopropanol precipitation. cDNA synthesis was performed using SMARTScribe™ Reverse Transcriptase (Takara, Inc., Kusatsu, Japan), and qPCR was performed on a CFX96 real-time thermal cycler (Bio-Rad). See Appendix A for the primer sequences.

### 2.13. Cell Cycle Analysis

The cells were cultured in DMSO or DOX, as indicated, collected and washed with cold PBS, and then fixed in ice-cold 70% EtOH for 30 min at 4 °C. The cell pellets were washed in staining buffer (PBS no Ca/Mg, 3% FBS, +1 mM EDTA), then incubated in staining buffer with 80 μg/mL Propidium Iodide and 0.125 mg/mL RNAse A (Thermofisher) for 40 min at 30 °C. The samples were run on a BD LSRII (BD Biosciences) and analyzed using FlowJo software (version 10.8.2).

### 2.14. Statistical Analysis

Statistics were performed using GraphPad Prism 5.02, utilizing the Student’s unpaired t-test for two means, one-way ANOVA with Tukey post-test for multiple means, or Kaplan–Meyer survival analysis with Mantel–Cox tests, as indicated. The significant *p*-values are listed or noted as * *p* < 0.05, ** *p* < 0.01, or *** *p* < 0.001. Error bars represent standard error means.

## 3. Results

### 3.1. AKT2 Depletion Impairs Cell Migration and Invasion in Human Melanoma Cells

It is well-known that AKT phosphorylation, a surrogate marker for active AKT, increases with disease stage in clinical samples [35], but there is a paucity of information regarding the relative contributions of individual isoforms to disease progression. In order to investigate this, we first characterized total AKT phosphorylation (Ser473 and Thr308) across a panel of human melanoma cell lines. Immunoblotting revealed a wide range of AKT phosphorylation levels and several cell lines that showed significant AKT2 phosphorylation (Appendix A). Furthermore, as expected, there was an inverse correlation between PTEN expression and total AKT phosphorylation (Appendix A).

In order to investigate the possible contribution of AKT isoforms to metastatic potential in human cell lines, we generated luciferized doxycycline-inducible shRNA hairpins to AKT1, AKT2, and AKT3 [34], as well as a nontargeting hairpin (shNT) that efficiently reduced protein expression in the majority of melanoma cell lines from our panel, (Figure 1A,B and Appendix A) without affecting viability (Figure 1C). We focused on cell lines in which there was significant AKT2 phosphorylation and robust knockdown efficiency (WM1799, UACC903, and WM455) and then sought to test the individual functions contributing to metastasis in these cells. By using a conventional in vitro wound healing assay, it was found that only AKT2 depletion and not AKT1 or AKT3 depletion inhibited cell migration in three different human melanoma cell lines (Figure 1D,E and Appendix A). Next, we utilized a complementary trans-well assay and observed that AKT2 depletion reduced cell migration in response to a serum gradient in multiple cell lines (Appendix A). Further, invasion through Matrigel was impaired by AKT2 depletion in the same human melanoma cell lines (Figure 1F,G and Appendix A). This reduction was not due to defects in cellular proliferation, as AKT2 did not impair cell proliferation in any cell line, as assessed by cell counting with trypan blue exclusion (Appendix A). When taken together, these data suggest that AKT2 depletion in vitro impairs the functions required for melanoma metastasis, such as migration and invasion.

### 3.2. AKT2 Depletion Restricts Anchorage-Independent Growth In Vitro and In Vivo

Anchorage-independent growth is required for the growth of metastatic cells; therefore, we interrogated the role of AKT2 in this process. By focusing our efforts on WM1799 cells, we seeded shAKT2 transduced WM1799 cells in soft agar, overlaid with either DMSO- or doxycycline-containing media to knock down AKT2. We observed that AKT2 depletion reduced the colony number (Figure 2A,B), with only a minor trend in decreased colony size (Figure 2C), indicating that AKT2 depletion limits the ability of cells to grow in a 3D culture. Next, we assessed the effect of AKT2 depletion on the growth of WM1799 cells as subcutaneous tumors in vivo, which similarly requires anchorage-independent growth in early tumorigenesis. We injected WM1799 shAKT2 cells subcutaneously into immunodeficient NOD/SCID mice (to exclude changes in melanoma cell growth due to an adaptive immune response) and allowed palpable tumors to form before transitioning a subset of these mice to doxycycline chow (Figure 2D). AKT2 knockdown significantly slowed tumor growth relative to the control mice fed with regular chow (Figure 2E), but tumors eventually grew in both groups despite persistent AKT2 knockdown (Figure 2F) in the majority of tumors. Further, we observed no change in total AKT phosphorylation across those tumors in which AKT2 was depleted, suggesting AKT1 or AKT3 activity may compensate over time to promote tumor growth.

### 3.3. AKT2 Depletion Delays Metastatic Onset and Extends the Survival of Melanoma-Bearing Mice

In order to determine whether AKT2 is important for the process of metastatic seeding and metastatic nodule growth, we performed a tail vein metastasis assay in which WM1799 shAKT2-Luc melanoma cells were allowed to seed the lungs. This was followed by initiating AKT2 depletion 24 h later by switching some mice to doxycycline-containing chow compared to mice fed with regular chow (Figure 3A). By performing immunoblot analysis using lysate derived from isolated pulmonary metastases, we confirmed that stable and robust AKT2 depletion occurred only in the doxycycline-fed mice (Figure 3B). By using weekly imaging, we found that the control mice fed regular chow displayed advanced metastatic disease at 6 weeks, with tumors observed at multiple sites; however, the mice fed doxycycline chow displayed only occasional tumor nodules at 6 weeks, albeit with significantly reduced size and frequency compared to the control mice (Figure 3C,D). While AKT2 depletion after metastatic seeding significantly improved overall survival in the doxycycline chow-fed mice versus regular chow mice, the doxycycline-fed mice did succumb to eventual metastatic disease, suggesting that AKT2 reduction only delayed the growth of the metastatic lesions, and other AKT activity might compensate over the long term (Figure 3E).

### 3.4. AKT2 Phosphorylation Occurs in Metastatic Mouse Melanoma Lesions

In order to study AKT2 in the context of metastatic murine melanoma, we generated an aggressive murine melanoma cell line, SM1-750, which exhibits high metastatic potential. The SM1 cell line was previously derived from melanomas arising in a BRAFV600E mouse and is tumorigenic in syngeneic mice [36]. In our hands, the subcutaneous tumor formation by SM1 cells in the C57BL6/J strain occurred in only a small subset of injected mice. In order to increase the tumor “take rate”, we passaged the SM1 cell line in tumor-forming C57BL6/J mice through several rounds of injection and tumor formation, resulting in the isolation of the SM1-750 line, which displays a nearly 100% take rate. Subsequently, the SM1-750 line was engineered to express luciferase, and the maintenance of the ability of these cells to form tumors was confirmed (Figure 3F and Appendix A).

The SM1 cells exhibited significant AKT1 and AKT3 phosphorylation but nearly undetectable AKT2 phosphorylation and the newly derived SM1-750 cells showed further elevated phosphorylation in terms of only AKT1 and AKT3. The SM1-750 cells were injected into the tail vein of syngeneic mice, and the injected mice were imaged after luciferin injection to visualize metastatic progression (Figure 3G). The mice readily developed metastatic lesions, which could be visualized in the lungs, brain, and liver (Figure 3H). Spontaneous brain metastases are relatively rare in murine melanomas and have previously been linked to AKT1 activation [27]. Interestingly, the immunoblotting of AKT isoform phosphorylation in discrete lung tumor metastases from individual mice revealed that AKT2 activation (indicated by S474 phosphorylation) dramatically increased when compared to that of primary tumors (Figure 3I). We further found that AKT2 upregulation in metastatic lesions was not due to adaptive immunity-dependent selection, as we also observed this in the SM1-750 metastases isolated from immune-deficient Rag2^−/−^ mice (Appendix A).

We then sought to characterize AKT isoform phosphorylation in spontaneously arising primary melanomas relative to murine melanomas with metastatic potential. We previously developed a mouse model of BRAF^V600E^-driven spontaneous melanoma, in which both melanocyte-targeted human BRAF^V600E^ co-operates with tumor suppressor p19^ARF^ loss (hereafter referred to as Arf^−/−^) to facilitate melanoma formation and AKT phosphorylation is observed in tumors but not normal skin [37]. The tumor suppressor Arf is commonly lost in human melanomas, and melanoma penetrance in our model increased on an Arf^−/−^ background [38]. We isolated the cell lines from the spontaneously arising primary tumors in these mice and investigated their isoform-specific phosphorylation patterns (Tufts University Mouse Melanoma, TUMM, see Appendix A). This analysis revealed ubiquitous total AKT phosphorylation on the activating residue serine 473 in the TUMM cell lines, as expected, as well as readily detectable AKT1 phosphorylation, with rare AKT3 phosphorylation and almost no AKT2 phosphorylation, which is in line with our observations of SM1-750 cells (Appendix A). Further, neither the TUMM nor SM1-750 cells showed AKT2 phosphorylation in low-attachment cell culture plates or in the primary tumors produced following subcutaneous injection into NOD/SCID mice (Appendix A), indicating that while AKT1 or AKT3 phosphorylation can drive melanoma cell growth and primary tumor formation, the activation of AKT2 correlates strongly with an ability to grow in the metastatic niche.

### 3.5. Prophylactic AKT2 Depletion Prevents Metastatic Cell Seeding

In order to determine if AKT2 plays a role in cellular migration and invasion, as well as in metastatic progression that is further extended to initial metastatic seeding, we pre-treated the WM1799 shAKT2 cells with doxycycline-containing media (compared to DMSO-containing media alone) for 72 h prior to tail vein injection into the NOD/SCID mice; we then maintained the mice on doxycycline-containing food or regular chow for 6 weeks. Metastatic progression was then monitored by using luminescence (Figure 4A). The mice maintained on regular chow displayed luminescent tumor nodules within 3 weeks, which progressed to advanced metastatic disease by 6 weeks, including multiple distant metastases under the forelimbs, the head, neck, and mesenteric lining (determined at autopsy, not shown), which is consistent with lymphatic dissemination (Figure 4B,C). Contrary to the control mice, those mice that received the shAKT2 cells and doxycycline chow remained healthy, with no detectable tumors after 6 weeks (Figure 4B,C). Additionally, when maintained on doxycycline chow, these mice were protected from detectable metastasis, even for up to 12 weeks (Figure 4D,E). These findings suggest that AKT2 is either required for metastatic seeding or the growth of seeded cells but do not distinguish whether AKT2 KD cells were eliminated from circulation or were simply latent in the mice fed with the doxycycline chow. In order to address this question, we asked if tumors would emerge after the removal of doxycycline chow. A subset of mice injected with WM1799 shAKT2 Luc cells and fed doxycycline chow were switched to regular chow after the first 6 weeks, at which time they still did not have detectable metastases. The mice were monitored weekly, and after an additional 6 weeks, none of the mice that were removed from doxycycline chow developed metastases (Figure 4F,G). These results suggest that the prophylactic targeting of AKT2 impairs the seeding of invasive cells in the metastatic niche, fully preventing metastatic formation rather than restraining the growth of dormant but intact metastatic tumor cells.

### 3.6. AKT2 Deletion Impairs Melanoma Migration, Invasion, and Metastasis

In order to confirm these findings with a more robust depletion of AKT2, we utilized isoform-specific CRISPR/Cas9 knockout in WM1799 cells for which we previously confirmed stable isoform-specific knockout (Figure 5A and Appendix A) [33]. We first characterized the properties of AKT2 knockout (KO) WM1799 cells compared to nontargeting (NT) cells in vitro, as was carried out for the inducible knockdown cell lines (Figure 1 and Figure 2). As expected, and in line with our results using inducible knockdown, we found that the AKT2 KO cells had impaired wound healing (scratch assay), migration (trans-wells), and Matrigel invasion when compared to the NT cells (Figure 5B,D), which are all indicative of impaired metastatic properties. In order to test this, we further engineered NT and AKT2 cells to express luciferase and injected them into the tail veins of NOD SCID mice (Figure 5E). At 8 weeks post-inoculation, the NOD SCID mice that received AKT2 KO cells had significantly lower metastatic burden than those injected with NT cells (Figure 5F,G). Further, the mice injected with AKT2 KO cells showed a survival benefit relative to NT-injected mice (Figure 5H), which is consistent with the previous results, indicating that AKT2 depletion delays the onset of metastatic disease and overall survival.

### 3.7. AKT1 Deletion Impairs Primary Tumor Formation

We next sought to test if AKT2 knockout impaired tumor initiation when compared to AKT1 and AKT3 deletion, as our CRISPR-edited WM1799 cells exhibited more robust AKT2 deletion compared to the inducible knockdown lines, which did not impact tumor initiation. We injected each isoform-specific knockout into the flanks of the NOD/SCID mice and monitored tumor formation and growth using caliper measurements (Appendix A). Interestingly, we found that while the AKT2 and AKT3 KO WM1799 cells developed tumors similarly to NT cells, only the AKT1 KO cells had significantly delayed tumor growth (Appendix A).

In order to further investigate the impact of the genetic loss of each AKT isoform on melanoma progression, we crossed melanoma-prone BRAF^V600E^; Arf^−/−^ mice with AKT isoform knockout mice [23] to generate a BRAF^V600E^; Arf^−/−^; AKT^−/−^ compound mutant mouse (Breeding scheme; Appendix A). As the BRAF^V600E^; Arf^−/−^ mice demonstrate significant tumor formation and overall reduced survival [38], we analyzed the contribution of each AKT to long-term survival. Again, AKT1 loss in the context of oncogenic BRAF and Arf loss provided a survival benefit for these melanoma-prone mice, which was not observed for AKT2 or AKT3 loss (Appendix A). This is in line with our data showing only strong AKT1 phosphorylation in primary mouse tumors, distinguishing the role of AKT2 in metastatic seeding from the role of AKT1 in primary murine melanoma formation.

### 3.8. AKT1 Knockdown Impairs Cellular Proliferation and Anchorage-Independent Growth

In order to investigate why AKT1 but not AKT2 impaired primary tumor growth, we analyzed cellular proliferation in our inducible AKT1 and AKT2 knockdown melanoma cell lines. We found that in the WM1799, UACC903, and WM1158 cells, only AKT1 but not AKT2 knockdown impaired cellular proliferation after 4 days in culture with doxycycline-containing media (Appendix A). We went on to perform cell cycle analysis on the shWM1799 cells and discovered that doxycycline-treated AKT1 knockdown cells exhibited increased G1 arrested cells and fewer G2/M phase cells, whereas the NT and AKT2 knockdown cells showed no cell cycle defects in the presence of doxycycline (Appendix A). The AKT1 knockdown cells further showed significantly reduced BrdU incorporation in the presence of doxycycline when compared to the NT and AKT2 cells (Appendix A), confirming a role for AKT1 in cellular proliferation that is in line with the existing literature [17] and one mechanism of delayed primary tumor formation.

We further tested whether AKT1 played a role in anchorage-independent growth, as the AKT1 KO WM1799 cells showed not only delayed tumor formation but also a reduction in tumor growth. Indeed, the knockdown of AKT1 reduced colony formation in soft agar with a trend in decreased colony size (Appendix A). We went on to test this observation in vivo by implanting shAKT1 WM1799 cells into NOD/SCID mice and switching the mice to doxycycline-containing chow compared to regular chow (Appendix A). The mice on the doxycycline chow had significantly reduced tumor growth compared to the regular chow-fed mice, together showing that AKT1 drives tumor cell proliferation, whereas AKT1 and AKT2 both contribute to anchorage-independent growth.

### 3.9. AKT2 Depletion Inhibits EMT and Impairs Glycolysis through PDHK1 Activity in Melanoma Cells

In order to investigate the possible mechanisms whereby AKT2 could support metastatic growth and survival, we first interrogated the impact of AKT2 depletion on the epithelial-mesenchymal transition (EMT), an early step in the acquisition of invasive and metastatic capability [39]. Only AKT2 knockdown but not AKT1 or AKT3 knockdown in WM1799 cells reduced the expression of pro-metastatic transcription factors ZEB1 and Snail and the matrix-remodeling enzyme MMP2, which is concomitant with a trend in increased E-cadherin expression (Figure 6A). Additionally, only AKT2 knockdown decreased the expression of TEA domain (TEAD) genes (Figure 6A), which are invasion-associated genes previously implicated in melanoma metastasis [40]. These data are consistent with an isoform-specific role for AKT2 in programming melanoma cells for EMT transition and subsequent metastasis.

It has long been appreciated that tumor cells undergo metabolic reprogramming during tumor progression and especially during metastasis, with an increased reliance on glycolysis over oxidative phosphorylation, a phenomenon known as the Warburg effect [41]. In order to determine whether AKT2 knockdown directly disrupts glycolysis, we employed a Seahorse glycolytic rate assay (GRA), which quantitatively measures the proton efflux rate (PER) in real time under standard conditions disruptive to mitochondrial metabolism. Using shNT or shAKT2 WM1799 cells, we performed the glycolytic rate assay after 48 or 72 h of incubation with DMSO- or DOX-containing media (Figure 6B). A comparison of basal glycolytic metabolism in shNT and shAKT2 cells revealed that AKT2 knockdown suppressed basal glycolytic metabolism (Figure 6B,C). After 72 h of AKT2 knockdown, glycolysis that was independent of mitochondrial respiration was also suppressed since Rot/AA injection was unable to increase PER (Figure 6B). Compensatory glycolysis was also significantly reduced in the AKT2 knockdown cells compared to the DOX-treated WM1799 shNT cells (Figure 6C). Together, these results strongly suggest that selective AKT2 knockdown inhibits glycolysis in WM1799 human melanoma cells.

In order to explore the potential mechanisms underlying the observed AKT2-dependent glycolytic defect, we evaluated the impact of AKT2 suppression on key enzymatic regulators of glycolysis. Pyruvate dehydrogenase kinases (PDKs) are critical regulators of the pyruvate dehydrogenase complex (PDC) through inactivating phosphorylation events, with subsequent decreases in TCA cycle activity shifting towards anaerobic respiration [42]. Of the four PDK isoforms (PDHK1-4), PDHK1 is the most frequently studied in the context of tumorigenesis and has previously been shown to be a direct target of AKT2 in prostate adenocarcinoma cells [42]. We first confirmed that PDHK1 was an AKT target in WM1799 cells by performing the immunoprecipitation of PDHK1 in the presence or absence of the AKT inhibitor MK2206 and probing for the phosphorylation of the AKT consensus site, showing AKT-specific phosphorylation on isolated PDHK1 that was significantly decreased by MK2206 treatment (Figure 6E). In order to confirm the isoform specificity of PDHK1 phosphorylation, we performed PDHK1 immunoprecipitation in our CRISPR KO WM1799 cell lines and found that only AKT2 KO cells showed decreased PDHK1 phosphorylation when compared to NT, AKT1, or AKT3 KO cells (Figure 6F). Further, we found that the majority of the immunoprecipitated PDHK1 was in the dimerized state, which occurs when PDHK1 is dephosphorylated and, therefore, inactive [43], supporting that the activity of AKT2 is specifically required for PDHK1-induced glycolytic activity.

We further tested the effect of AKT2 knockdown on PDHK1 levels and found that while shAKT2 cells treated with doxycycline resulted in similar levels of PDHK1 (Figure 6G), a major target of PDHK1 kinase activity, pyruvate dehydrogenase E1 component subunit alpha (PDEH1α), had significantly lower PDEH1α phosphorylation in only those shAKT2 cells treated with doxycycline for 72 h. Decreased PDEH1α phosphorylation as a result of impaired PDK1 activity would be expected to shift metabolic flux away from glycolysis back to the TCA cycle, which is one mechanistic explanation for the loss of glycolytic activity in shAKT2 cells. Overall, our findings are consistent with a role for AKT2 in the regulation of not just EMT but also glycolytic versus aerobic respiration, promoting melanoma invasiveness and metastatic activity.

## 4. Discussion

Despite clear evidence that the PI3K/AKT pathway is important for tumor progression and metastasis in melanomas, the effective therapeutic inhibition of the AKT pathway has been challenging [10]. A potential pitfall in targeting the AKT pathway may be the strategy of pan-inhibition despite strong evidence of AKT isoform specificity in cancer. This study sought to investigate the contribution of AKT isoforms to melanoma initiation and metastasis, utilizing both murine and human melanoma models.

Although it is well-known that AKT activation drives tumor progression and metastasis in melanoma [44,45,46,47], the contributing AKT isoform(s) were not determined. Our data support an isoform-specific role for AKT2 in melanoma metastasis, with a shared role for AKT1 and AKT2 in tumorigenesis. Initially, we crossed mice with BRAF-driven murine melanoma previously developed in our lab [38] with mice lacking different AKT isoforms [23] and monitored the mice derived from these crosses for melanoma development. We observed that AKT1 loss extended overall survival in melanoma-prone mice, while AKT2 and AKT3 loss did not (Figure 1C). This suggests that AKT1 promotes melanoma initiation and growth, which is consistent with a well-described role for AKT1 in tumor promotion [15,21] and cell proliferation and survival [17]. This is also consistent with our observation that tumor-derived cell lines consistently show robust phosphorylation of AKT1 when compared with other AKT isoforms (Figure 3F and Appendix A), suggesting that AKT1 but not AKT2 or AKT3 plays a critical role in melanoma initiation. One caveat to these findings is that 20% of BRAF^V600E^/Arf^−/−^ mice die for reasons unrelated to melanoma [38], and determining the exact cause of death was not technically feasible in our initial studies. Additionally, while the effect of AKT1 on survival implies PI3K-AKT activation in this model, we did not explicitly test the extent of this within the resulting murine tumors. As such, melanoma-free survival or the exclusion of animals that both lacked melanoma and also died for reasons unrelated to melanoma would more accurately reflect the relative contributions of AKT isoforms to melanoma initiation and tumor promotion. However, because early primary melanomas pose a relatively low clinical risk, we focused our continuing studies on metastatic disease progression, which still presents the greatest treatment challenge for patients.

In order to address the potential involvement of AKT isoforms in metastasis, we chose to examine how the loss of AKT isoforms affects the metastatic potential of a melanoma cell line we established from a BRAF^V600E^/Arf^−/−^ mouse melanoma. The cell line was engineered to express luciferase, and following IV inoculation in syngeneic immunocompetent mice, it could be monitored by bioluminescence. By using this approach, we discovered that increased phosphorylation of AKT2 was present in metastatic lesions despite the lack of such phosphorylation in primary tumors and parental cell lines, suggesting that AKT2 activation may facilitate the metastatic seeding, survival, and/or growth of the cells seeded in metastatic sites. We have previously reported a low but significant rate of spontaneous lung metastasis in the BRAF^V600E^/Arf^−/−^ melanoma model [38]. Based on this observation, future studies could use this model to determine whether the AKT isoforms differentially control the rate of metastatic melanomas. This could be addressed by monitoring the metastatic burden of BRAF^V600E^/Arf^−/−^/AKT^−/−^ mice lacking individual AKT isoforms.

The preceding data suggest that AKT2 differentially promotes the metastatic potential of melanomas, but to address the mechanism, we utilized a set of human melanoma cell lines driven by mutation in BRAF, as well as by PTEN loss. The functional loss of the tumor suppressor PTEN leading to AKT activation occurs in a large proportion of melanomas [7], and frequently co-occurs with oncogenic BRAF mutations [26]. However, the isoform specificity of AKT in the context of PTEN loss has remained largely unexplored. Therefore, using several BRAF^V600E^ metastatic human melanoma cell lines with PTEN loss and robust AKT phosphorylation, we investigated the consequence of isoform-specific knockdown on cell migration and invasion. Our results show that AKT2 depletion consistently inhibits invasive and migratory behaviors, while AKT1 depletion reduces cellular proliferation and tumor growth. We postulated that AKT2-dependent invasion and migration were regulated through the AKT2-specific transcriptional changes in EMT-related genes E-cadherin, ZEB1, and Snail (Figure 6A). These data are consistent with findings in breast cancer, in which AKT2 but not AKT1 promotes cell migration and EMT [21,22,48,49] and the well-recognized role of EMT in metastatic promotion [50,51,52]. While our studies using shRNA-mediated AKT depletion used only one hairpin per isoform, the similar phenotypes between shRNA depletion and CRISPR knockout cells support the conclusion that impaired metastatic capability depends more specifically on AKT2.

While a consistent and explicit role for AKT3 was not identified in our study, this may be partially explained by AKT3-specific knockdown efficiency, which varied greatly among the cell lines (Appendix A). AKT3 amplification is known to occur in melanoma [26] but was not characterized across our panel and could attenuate knockdown efficiency if present, which is consistent with reports in the literature showing robust AKT3 phosphorylation in melanoma [24]. Nevertheless, our finding that AKT2 knockdown greatly attenuates the metastatic potential of human melanoma cell lines is consistent with previous data, suggesting that PHLPP1 inhibits melanoma metastasis by suppressing the phosphorylation of AKT2 and AKT3 but not AKT1 [25].

Once cells become migratory and invasive, extravasation and anchorage-independent growth are two necessary requirements for metastatic colonization [50]. Previous work in PTEN-deficient prostate tumors demonstrated their dependence on AKT2 for both maintenance and survival, but AKT1 in the same tumors was dispensable [53]. Our xenograft studies suggest that PTEN-deficient melanomas are initially sensitive to AKT2 inhibition but ultimately do not depend on AKT2 over the long term, given our observation of delayed tumor outgrowth in the AKT2 knockdown mice, as well as unmitigated tumor growth of AKT2 KO cells (Appendix A). In PTEN null melanomas and in the human cell lines in our study, a key difference is significant AKT1 phosphorylation, which could compensate to drive tumor cell growth and survival, given the cellular sensitivity to AKT1 depletion in cell proliferation and tumor growth that we observed. We hypothesize that potential compensation by AKT1 might explain why the CRIPSR KO AKT2 cells did not recapitulate the decreased subcutaneous tumor growth or the full prophylactic protection observed for the shRNA AKT2 KD cells, although we have not explicitly tested this hypothesis. Despite the apparent inability of AKT1 to fully support metastasis in AKT2-depleted human melanoma cells, hyperactive AKT1 contributes to the metastatic potential of the mouse SM1-750 model. By utilizing an autochthonous mouse model of melanoma, Kircher and colleagues showed that hyperactive, ectopic AKT1(E17K) resulted in an enhanced level of brain metastases and reduced overall survival compared to hyperactivating mutations in AKT2 or AKT3, mediated through a FAK an AKT1-specific substrate [32]. Indeed, we observed a significant increase in the phosphorylation of both AKT1 and AKT3 after passage in vivo to generate the brain metastatic-competent SM1-750 cell line (Figure 3F), in addition to increased AKT2 phosphorylation despite the presence of intact PTEN protein (Appendix A). Whether this increased phosphorylation is due to hyperactivating mutations arising in AKT1 remains to be determined. Interestingly, Kircher and colleagues also showed that mice with hyperactive AKT2 but not hyperactive AKT3 mutations still developed brain metastases, albeit to a lesser extent than AKT1. We are currently unable to perform AKT2 overexpression in our model systems due to the inability to interpret isoform-specific effects in the context of constitutively active AKT; however, future studies will aim to utilize this approach to test if AKT2 overactivity increases metastatic capability. These observations, when taken together with our data, suggest that AKT2 promotes melanoma metastasis in murine model systems, and further studies may elucidate the mechanistic differences in metastatic promotion between AKT isoforms in murine melanomas.

In order to understand the potential stages of metastatic dissemination for which AKT2 may be most critical, we explored extravasation from blood vessels and tumor cell proliferation at seeded distant sites as two candidate stages for the study of metastasis. The data presented in this report provide strong evidence that the key limiting event regulated by AKT2 was tumor cell extravasation. By pre-incubating cells with doxycycline to knock down AKT2 and then injecting them into the tail vein of mice, we observed the complete lack of metastatic dissemination despite the presence of phosphorylated AKT1 and AKT3. We ruled out the possibility that resident cells were dormant at distant metastatic sites by removing mice from doxycycline chow at 6 weeks. No tumors appeared after an additional 6-week observation period, suggesting that the disseminated cells may have been removed from circulation. It is also possible that 6 weeks was not sufficient time or that the dormant cells were too small to observe by our methods, as melanoma cells disseminating to the brain have been known to exist as single cells or small clusters barely visible by histology that can then regrow when the conditions are optimal [54]. Nevertheless, when cells expressing AKT2 were injected into the tail vein, and then mice were fed doxycycline chow to knock down AKT2 24 h later, there was a partial inhibition of metastasis. However, these mice were not fully protected from metastatic disease, unlike the mice inoculated with AKT2-depleted cells, suggesting AKT2 activity may be most impactful at the stage of extravasation, and its effect on tumor growth at the metastatic site was moderate. These results were phenocopied with AKT2 KO cells, in which AKT2 KO provided a survival benefit, perhaps by partially inhibiting the onset of metastatic disease.

Lastly, we investigated the mechanisms whereby AKT2 might mediate pro-metastatic behaviors, hypothesizing that AKT2 KD cells might be glycolytically impaired. Further, the relationship between impaired glycolysis and impaired EMT is well-established in the literature [55,56]. AKT2-specific roles in maintaining glucose homeostasis are well-documented [18], and metabolic rewiring in melanoma can facilitate metastatic dissemination [57]. We observed that AKT2 KD suppressed basal glycolytic metabolism and reduced compensatory glycolysis. In malignant glioma, the AKT2-specific phosphorylation of PDHK1 at Thr346 was shown to increase the phosphorylation of PDHE1α. This interferes with the entry of pyruvate into the TCA cycle, resulting in the stimulation of glycolysis, the maintenance of cell proliferation, and the inhibition of autophagy and apoptosis during severe hypoxia [42]. Earlier studies have shown that AKT2 selectively promotes the expression of miR-21 during hypoxia and renders cells resistant to hypoxia-induced cell death [58] Importantly, recent studies have shown that miR-21 promotes glycolysis by targeting pyruvate dehydrogenase A1 (PDHA1) [59] Therefore, AKT2 may inhibit the activity of pyruvate dehydrogenase by increasing the activity of PDHK1 and inhibiting the expression of PDHA1. In agreement with a significant role for AKT2-mediated regulation of PDHK1 and subsequent PDHE1α activity, we observed reductions in PDHK1 activity as well as phosphorylated PDHE1α with AKT2 KD. When taken together, the totality of our data suggests the AKT2-mediated regulation of PDHK1 and other components of pyruvate metabolism are important in both normal and hypoxic conditions, although the specific regulation may differ. Future studies should further define the role of PDHK1 in glycolytic maintenance that mediates melanoma metastasis.

## 5. Conclusions

In summary, by using genetically engineered human cell lines and novel syngeneic mouse models, we reveal multiple roles for AKT2 in melanoma metastasis. Indeed, as AKTs are central pleiotropic signaling hubs [60], it is not surprising that the AKT2-mediated cellular changes facilitating melanoma metastasis are manifold, including glycolytic changes and the enhancement of factors promoting epithelial-to-mesenchymal transition. This study reinforces the need for improved development in terms of clinically relevant small molecules for the selective targeting of AKT isoforms as a melanoma therapy.

## Figures and Tables

**Figure 1 cancers-15-04958-f001:**
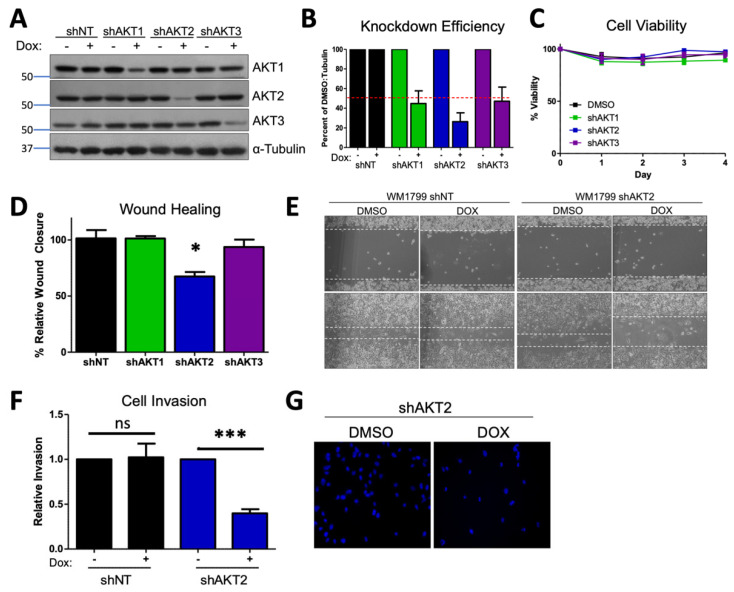
AKT2 depletion impairs migration and invasion in WM1799 human melanoma cell lines. (**A**) A representative immunoblot of WM1799 human melanoma cells after stable cell line generation. Dox-inducible AKT-isoform knockdown (shAKT1/2/3) or nontargeting hairpin (shNT) expressing cell lines are shown after 72 h of treatment with 0.5 ug/mL doxycycline. (**B**) Quantitation of KD efficiency from three independent experiments using ImageJ to quantify the intensity of the bands normalized to percent of DMSO-treated total protein and loading control. (**C**) The cell lines from (**A**) were grown in the presence of DOX or DMSO and collected at indicated times for cell counting with trypan blue exclusion to assess cell viability from two independent experiments. (**D**) Quantitation of wound closure in shWM1799 cells at 0 h and 16 h post-wounding, comparing DMSO- and DOX-treated cells from three independent experiments, with representative images at 4× magnification (**E**) of shNT or shAKT2 WM1799 cells treated with DOX relative to DMSO vehicle-treated cells. (**F**) Quantitation of invasion ability of control (NT) or AKT2 KD (shAKT2) cells with or without DOX treatment, as indicated from three independent experiments. (**G**) Representative images of DAPI-stained AKT2 KD cells on the underside of a Matrigel-coated membrane (20× magnification) following treatment with DMSO or DOX (left) from n = 3 total experiments. ns: no significant difference, *: *p* < 0.05, ***: *p* < 0.001.

**Figure 2 cancers-15-04958-f002:**
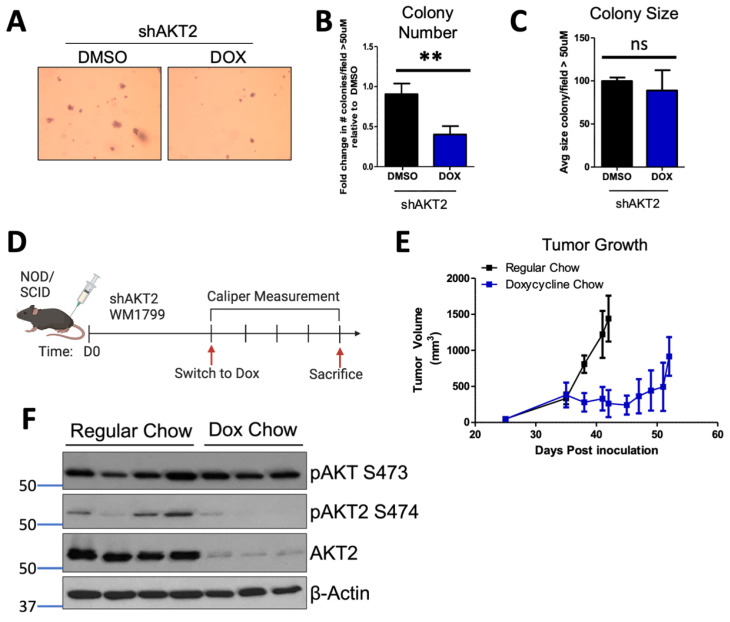
AKT2 depletion restricts anchorage-independent growth. (**A**) Anchorage-independent growth was assessed by seeding WM1799 AKT2 KD cells in soft agar and culturing in DMSO- or DOX-containing media for three weeks, after which the cells were fixed and stained with crystal violet (left; representative image at 4× magnification). (**B**,**C**) Colonies greater than 50 µM were counted and quantified using ImageJ from three independent experiments. (**D**) Schematic of experimental design to assess tumor growth potential. WM1799 shAKT2 cells were injected subcutaneously into NOD/SCID mice and allowed to form palpable tumors; then, the mice were randomized into groups receiving either regular or DOX-containing chow to induce AKT2 KD (n = 3–4 mice per group). (**E**) Tumor volumes were determined using calipers at the indicated times, and the mice were sacrificed when tumors reached 1500 m^3^ in size. (**F**) The tumors isolated from individual mice receiving either DMSO or DOX chow were subjected to immunoblotting for AKT2 protein and phosphorylation using beta-actin as a loading control. ns: no significant difference, **: *p* < 0.01.

**Figure 3 cancers-15-04958-f003:**
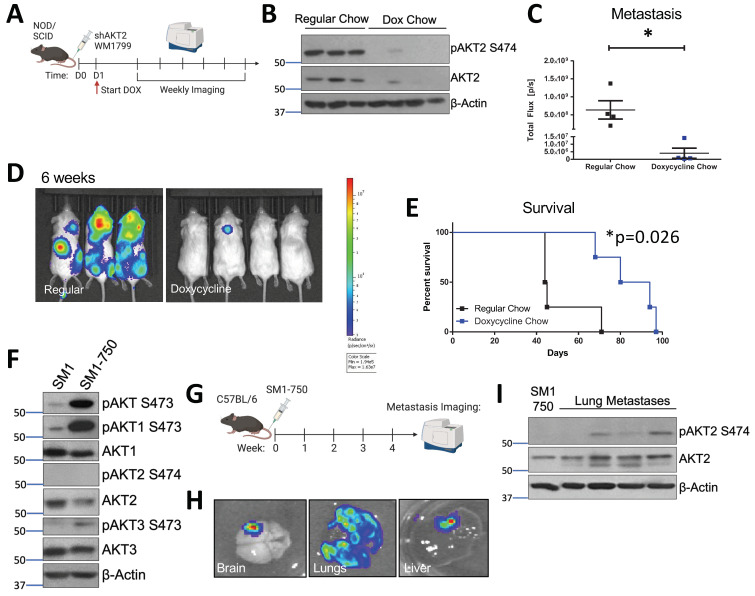
AKT2 depletion delays metastatic onset and extends the overall survival of melanoma-bearing mice. (**A**) Schematic representation of experimental design. Luciferized WM1799 cells expressing DOX-inducible shAKT2 hairpins were injected into the tail vein of NOD/SCID mice. The mice were started on DOX chow 1 d post-injection, and were maintained on regular chow or DOX chow and monitored weekly using IVIS SpectrumCT imaging. (**B**) Immunoblot analysis of total and phospho-S474 AKT2 (P-AKT2) levels in the metastatic tumors isolated from the mice fed regular or doxycycline chow at the time of death. (**C**) Quantification of mice bearing metastases at 6 weeks post-injection, and the representative images (**D**) of luminescence 6 weeks after cell injection in the mice fed doxycycline chow compared to regular chow (N = 3–4 mice per group). (**E**) Percent survival of mice bearing metastases. (**F**) SM1 and SM1-750 cell lines were established from spontaneous primary murine melanomas expressing BRAF^V600E^ and passaged in the C57Bl6/J mice; this was then analyzed by immunoblotting. (**G**) Schematic of experimental design of in vivo metastasis assays. SM1-750 cells expressing luciferase were injected into the tail vein of C57Bl6/J mice, and the tissues were analyzed using IVIS SpectrumCT imaging at 4 weeks post-injection. (**H**) The lungs, liver, and brains collected from the mice were imaged ex vivo, and the representative images from n = 4 mice are shown. (**I**) The lung metastases were excised from four individual mice (at least 3–4 nodules per mouse were combined) and homogenized to generate protein lysates; this was analyzed by immunoblotting. *: *p* < 0.05.

**Figure 4 cancers-15-04958-f004:**
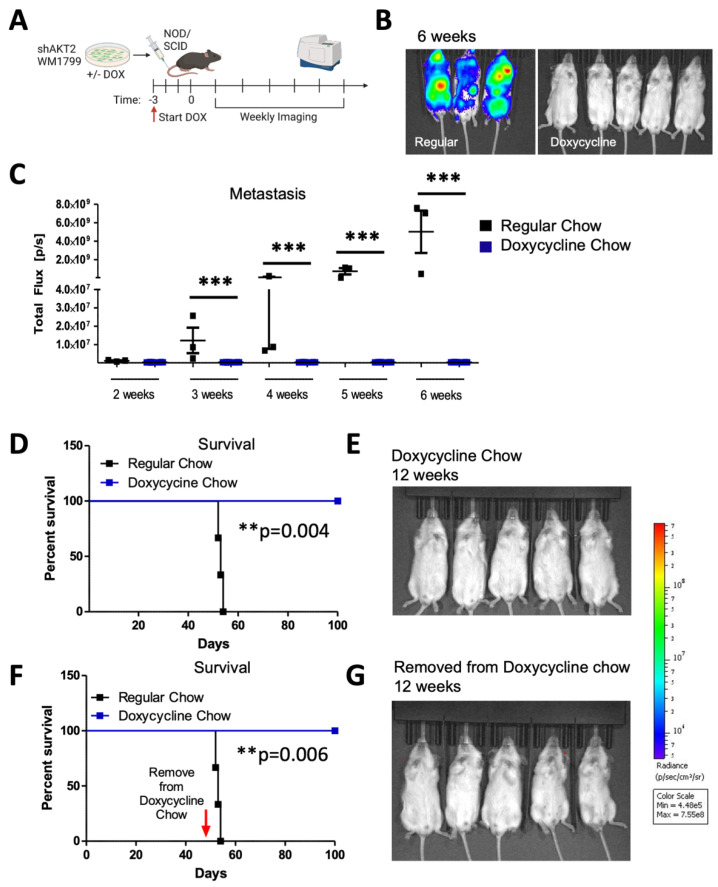
Prophylactic AKT2 depletion prevents metastatic cell seeding. (**A**) Schematic representation of experimental design. AKT2-depleted (3 days of DOX treatment) or control (DMSO treated) WM1799 shAKT2 Luc cells were injected into the tail vein of the NOD/SCID mice fed DOX chow or regular chow, respectively, for 3 days prior to injection (N = 3–5 mice per group). Mice were provided with regular or DOX chow and monitored weekly by IVIS SpectrumCT imaging. (**B**) Image of mice 6 weeks post-injection for DOX-treated versus regular chow mice. (**C**) Quantification of luminescence from mice in (**B**) using LivingImage software at indicated times. (**D**) Survival of regular-chow-fed (control) and DOX-fed mice injected with DMSO- or DOX-treated shAKT2 Luc cells, respectively (n = 3–5 mice per group). Mice were sacrificed when moribund. (**E**) Luminescence assayed at 12 weeks for mice injected with AKT2 KD cells and fed Dox chow. (**F**,**G**) A subset of mice treated as per D and E that were switched to regular chow after 6 weeks on DOX chow; these were assessed for the presence of metastases by IVIS SpectrumCT imaging after an additional 6 weeks and showed no evidence of progressing metastatic lesions (n = 3–5 mice per group). **: *p* < 0.01, ***: *p* < 0.001.

**Figure 5 cancers-15-04958-f005:**
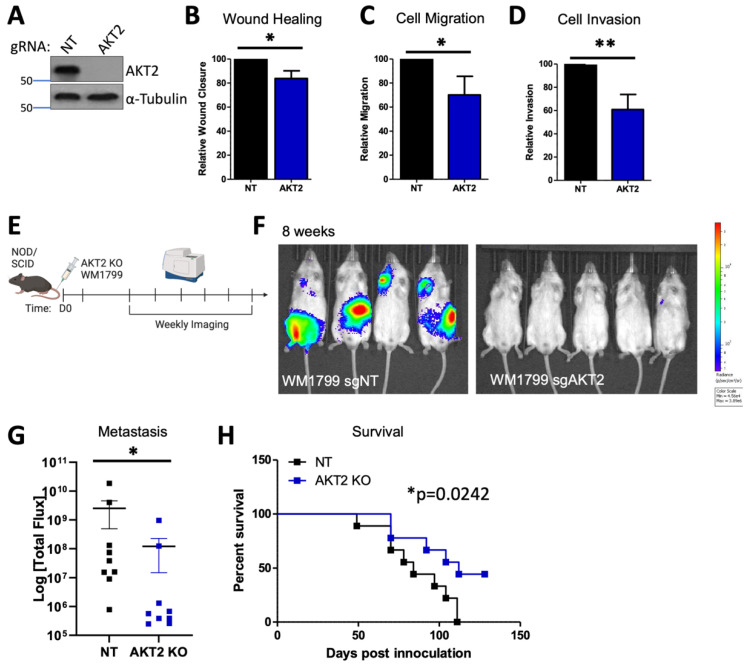
AKT2 Knockout impairs human melanoma cell migration, invasion, and metastasis. (**A**) Representative immunoblot of KO WM1799 cells for AKT2. (**B**–**D**) Ability of WT or AKT2 KO WM1799 cells in wound closure using scratch assay (**B**), cell migration using trans-well assay (**C**), or cell invasion through a Matrigel-coated membrane (**D**), all from n = 3 independent experiments. (**E**) Experimental schematic, in which AKT2 KO WM1799 cells were engineered to express luciferase and were injected into the tail veins of NOD/SCID mice; metastasis was monitored using IVIS SpectrumCT imaging; representative mice at 8 weeks are shown (**F**) with quantification (**G**). (**H**) Overall percent survival of mice injected with WM1799 AKT2 KO cells compared to NT cells. *: *p* < 0.05, **: *p* < 0.01.

**Figure 6 cancers-15-04958-f006:**
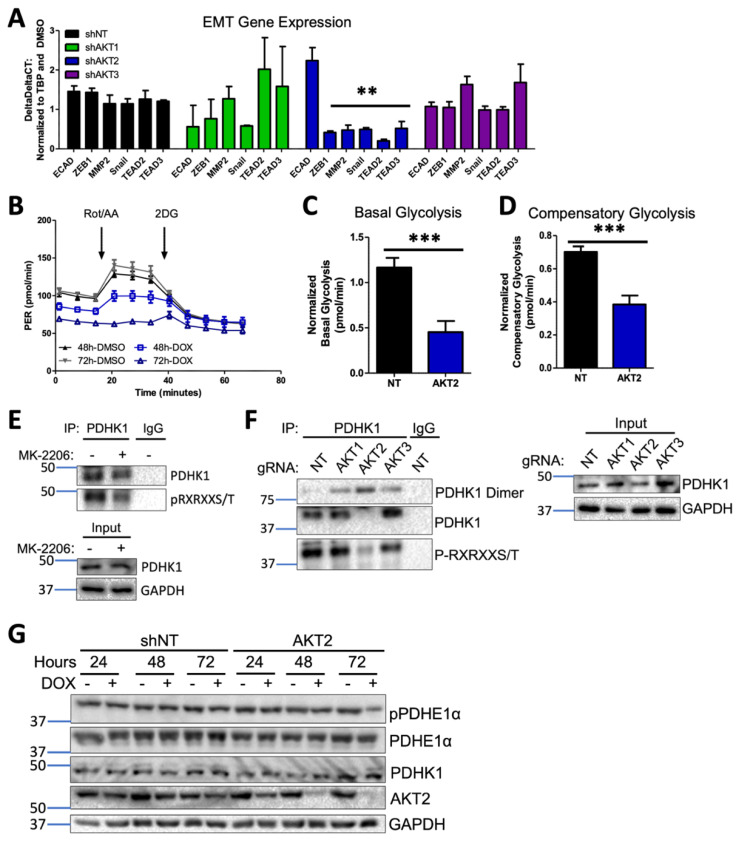
AKT2 depletion alters EMT and impairs glycolysis in melanoma cells. (**A**) RT-qPCR analysis of EMT and invasion-associated transcripts in WM1799 shNT, AKT1 KD, AKT2 KD, or AKT3 KD cells after 48 h of DOX treatment from three independent experiments. Expression levels were normalized to shNT cells. (**B**) Representative GRA plot shows the proton efflux rate (PER) of WM1799 shAKT2 cells cultured in DMSO- or DOX-containing media for 48 or 72 h, which was used to quantify basal glycolysis (**C**) and compensatory glycolysis (**D**) in DOX-treated WM1799 shNT cells from three independent experiments. (**E**) PDHK1 immunoprecipitation was performed after 24 h of MK-2206 treatment, and then immunoblotting was performed for AKT consensus site phosphorylation, with the input immunoblot for PDHK1 shown in the lower panel. (**F**) PDHK1 immunoprecipitation was performed on isoform-specific CRISPR KO WM1799 cells, then immunoblotting was performed for AKT consensus site phosphorylation. (**G**) Immunoblot analysis of PDHK1 expression or PDHE1a expression and phosphorylation in WM1799 shNT or AKT2 KD after treatment with DOX for indicated time points. ns: no significant difference. **: *p* < 0.01, ***: *p* < 0.001.

## Data Availability

Data are available within the article or upon request from the authors.

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
