# Peer review of "AKT2 Loss Impairs BRAF-Mutant Melanoma Metastasis"

_cancers, 2023, doi:10.3390/cancers15204958_

Round 1

Reviewer 1 Report

The manuscript by Dr. Philip W. Hinds and colleagues  described different Akt isoforms’ effects on melonoma development.  AKT2 specifically regulates melanoma cell metastasis 1 while AKT1 is involved in cellular proliferation and growth.This paper is well constructed and suitable to be published when  issues solved and questions in this manuscript are addressed:

1. In Figure1E, author should also compare with shNT in DMSO and Dox condition to exclude that the effects is due to Dox.

2. In Figure 2C, corresponding to Figure 2A, the colony size seems to change a lot.

3. In Figure 6E, Igg control should be set as a control.

4. In Figure 6F, why longer time treatment show the different expression of PDHK1 between shNT and shAKT2?

Author Response

The manuscript by Dr. Philip W. Hinds and colleagues  described different Akt isoforms’ effects on melanoma development.  AKT2 specifically regulates melanoma cell metastasis 1 while AKT1 is involved in cellular proliferation and growth. This paper is well constructed and suitable to be published when  issues solved and questions in this manuscript are addressed:

  1. In Figure 1E, author should also compare with shNT in DMSO and Dox condition to exclude that the effects is due to Dox.

Response: We would like to clarify that the quantitation includes comparison between DMSO and DOX treated cells, now stated in the figure caption. We also now show an updated panel 1E to include representative images for both the shNT and shAKT2 cell lines, in which the shNT cell line undergoes robust wound healing in the presence or absence of Dox treatment, whereas the shAKT2 cell line undergoes impaired wound healing only in the presence of Dox, as quantified in Figure 1D.

  1. In Figure 2C, corresponding to Figure 2A, the colony size seems to change a lot.

Response: While we agree that there is a minor trend in decreased colony size in the shAKT2 cell line with Dox treatment, over 3 independent experiments averaging the variable colony sizes we did not find statistically significant changes in colony size. The representative images show well the decreased colony number. We have updated the text in lines 263-264 accordingly to acknowledge this trend.

  1. In Figure 6E, Igg control should be set as a control.

Response: We have updated Figure 6E to include an IgG control, showing the specificity of the PDHK1 IP with the same phenotype of decreased AKT substrate phosphorylation.

  1. In Figure 6F, why longer time treatment show the different expression of PDHK1 between shNT and shAKT2?

Response: We agree that the original figure did appear to have differential PDHK1 expression between cell lines, however this was not a consistent finding, and we have updated Figure 6G with a new replicate of this experiment that shows relatively consistent PDHK1 expression with significantly reduces PDHE1α phosphorylation, as a result of the decreased PDHK1 activity. This replicate best represents the majority of the replicates overall, and we thank the reviewer for calling attention to the anomaly in the original figure.

Reviewer 2 Report

In this article, McRee et al. investigated how specific AKT isoforms contribute to the development and metastasis of melanoma. They found that while AKT2 is found to be dispensable for primary tumor formation, it enhances migration, invasion, and metastatic seeding. In contrast, AKT1 is crucial for melanoma initiation and cell proliferation. They proposed a mechanism whereby Blocking AKT2 activity hinders the glycolysis and diminishes the expression of genes associated with EMT in PTEN-deficient, BRAF-mutated human melanoma cells. The in vivo models show promising phenotypes; however, it would be beneficial to consider some revisions and improvements to enhance the overall quality of the manuscript.

1.     Please indicate sample size (n)/number of independent experiments in all the figure legends that involve quantification.

2.     All western blot data should be presented adhering to the general publication requirements. For example, band sizes should be labelled for all the images.  

3.     In Figure 1C, the authors claimed knocking down AKT isoforms did not affect cell viability by using Trypan Blue exclusion. However, Trypan Blue only indicates the integrity of the cellular membrane; other assays need to be done to exclude the possibility of senescence.

4.     In Figure 2D-2E, please clarify how exactly “palpable tumors” were defined and determined, and the timing of transitioning a subset of mice to doxycycline chow.  

5.     Figure 3-5 showed strong phenotypes, but not enough investigations into the potential mechanisms. If other AKT activity could compensate the effect from inhibition of AKT2 and given the complexity of the PI3K/AKT pathway, AKT2 might not be an ideal therapeutic target unless there is a tumor-cell-specific targeting strategy.

6.     In Figure 6E, it only proved that PDHK1 was an AKT target but did not specify which isoform, which Figure 6F only showed some functional correlation. Other than using MK-2206 which is a pan-AKT inhibitor as a negative control, perhaps cell lines with AKT1, AKT2, AKT3 knockdown/knockout can be used to narrow down which isoform PDHK1 binds to. A reverse IP followed by western blotting of AKT isoforms using PDHK1-knockout line could also shed a light on this question.

Parts of the abstract and introduction are a bit verbose. Can be edited to be more concise. 

Author Response

I

In this article, McRee et al. investigated how specific AKT isoforms contribute to the development and metastasis of melanoma. They found that while AKT2 is found to be dispensable for primary tumor formation, it enhances migration, invasion, and metastatic seeding. In contrast, AKT1 is crucial for melanoma initiation and cell proliferation. They proposed a mechanism whereby Blocking AKT2 activity hinders the glycolysis and diminishes the expression of genes associated with EMT in PTEN-deficient, BRAF-mutated human melanoma cells. The in vivo models show promising phenotypes; however, it would be beneficial to consider some revisions and improvements to enhance the overall quality of the manuscript.

  1. Please indicate sample size (n)/number of independent experiments in all the figure legends that involve quantification.

Response: All figure legends now include the number of mice or number replicates of experiments performed.

  1. All western blot data should be presented adhering to the general publication requirements. For example, band sizes should be labelled for all the images.  

Response: All western blot images now include molecular weight markers

  1. In Figure 1C, the authors claimed knocking down AKT isoforms did not affect cell viability by using Trypan Blue exclusion. However, Trypan Blue only indicates the integrity of the cellular membrane; other assays need to be done to exclude the possibility of senescence.

Response: While we acknowledge that Trypan blue only is a readout for membrane integrity and thus cell death, our previous publication has already interrogated the role of each AKT isoform in senescence (PMID: 35158840). In this paper, we show that CRISPR knockout of either AKT1, AKT2, or AKT3 does not result in senescence at baseline, as shown by a lack of β-Gal staining and SASP transcript expression. As such, we do not expect shRNA induced knockdown of any isoform to result in senescence.

  1. In Figure 2D-2E, please clarify how exactly “palpable tumors” were defined and determined, and the timing of transitioning a subset of mice to doxycycline chow.  

Response: Thank you for requesting clarification. The inoculated flank for these mice was palpated 3x weekly until a subcutaneous tumor was felt by light touch compared to the non-inoculated flank (roughly 200 mm3)  which occurred at roughly 3-4 weeks post inoculation. Before introducing doxycycline chow, tumors from all shRNA expressing cells became palpable at similar time frames; as such, once all mice had palpable tumors (after 3-4 weeks) then doxycycline chow was initiated. This is now clarified in the methods.

  1. Figure 3-5 showed strong phenotypes, but not enough investigations into the potential mechanisms. If other AKT activity could compensate the effect from inhibition of AKT2 and given the complexity of the PI3K/AKT pathway, AKT2 might not be an ideal therapeutic target unless there is a tumor-cell-specific targeting strategy.

Response: We thank the reviewer for the interesting thought, and agree that the complexity of the PI3K/AKT signaling pathway makes therapeutic targeting a challenge. However, as we show isoform specific effects of AKT2 knockdown on melanoma metastasis, EMT signature, and PDHK1 phosphorylation (new Figure 6F), we believe AKT2 specific inhibition remains a potential therapeutic approach. However, AKT2 specific inhibitors do not yet exist. We did not explicitly test the compensation of other AKT isoforms in the absence of AKT2, however we agree this might be a challenge to the durability of effective AKT2 therapeutic modulation (that is, resistance to AKT2i may result from eventual compensation by the other isoforms), and future studies will address this. We note that many effective targeted therapies have significant effects upon initial treatment, while later succumbing to a variety of cellular resistance mechanisms; nevertheless significant positive impacts on PFS and OS are observed with such agents.

  1. In Figure 6E, it only proved that PDHK1 was an AKT target but did not specify which isoform, which Figure 6F only showed some functional correlation. Other than using MK-2206 which is a pan-AKT inhibitor as a negative control, perhaps cell lines with AKT1, AKT2, AKT3 knockdown/knockout can be used to narrow down which isoform PDHK1 binds to. A reverse IP followed by western blotting of AKT isoforms using PDHK1-knockout line could also shed a light on this question.

Response: To address this, we performed PDHK1 immunoprecipitation from the isoform specific CRIPSR knockout WM1799 cells, and now show that only AKT2 knockout cells exhibit decreased PDHK1 phosphorylation compared to NT, AKT1, or AKT3 KO. Further, we show that in the AKT2 KO cells, a majority of the PDHK1 was in the dimerized, and therefore inactive state (new reference 43), further supporting that the activity of AKT2 is required for PDHK1 induced glycolytic activity. These data are shown in new Figure 6F, and discussed in lines 463-468 of the manuscript.

Reviewer 3 Report

This is a great work that I have not reviewed for a long time. The study investigated the distinctive roles of 3 Akt isoforms in melanoma and found that Akt1 was primarily responsible for tumor growth while Akt2 mediated its metastasis. With this knowledge, we can develop better therapeutic strategies to treat or prevent melanoma. Just a minor question, Did you ever try to time the phosphorylation of the three isoforms? For instance, using clinical specimens from different stages of patients or from different organs? That would make the conclusion even stronger. Well done.

Author Response

This is a great work that I have not reviewed for a long time. The study investigated the distinctive roles of 3 Akt isoforms in melanoma and found that Akt1 was primarily responsible for tumor growth while Akt2 mediated its metastasis. With this knowledge, we can develop better therapeutic strategies to treat or prevent melanoma. Just a minor question, Did you ever try to time the phosphorylation of the three isoforms? For instance, using clinical specimens from different stages of patients or from different organs? That would make the conclusion even stronger. Well done.

Response: We thank the reviewer for the kind comments on our work. We have not been able to perform any kinetic or spatial analysis on isoform specific AKT phosphorylation from patient samples, in part due to difficult in obtaining sufficient sample quantities to perform these studies. While we suspect that heterogeneity in tumor samples, as well as the ability of AKT isoforms to compensate for one another, would make the interpretation of these results difficult, future studies will aim to better understand the timing of isoform specific AKT phosphorylation across melanoma progression.

Reviewer 4 Report

In the manuscript by McRee et al., entitled “AKT2 loss impairs BRAF-mutant melanoma metastasis”, the authors have investigated the role of individual AKT isoforms in melanomagenesis and metastatic dissemination in BRAFV600E-driven melanoma cell lines and mouse models.  Using a panel of BRAFV600E-driven human or mouse melanoma cell lines, they claim that knockdown of AKT2 inhibits cell migration and invasion.  They performed tail vein injections of cells with reduced or ablated of AKT2 to assess the metastatic potential of AKT2 and claim that AKT2 depletion impairs metastasis.  However, they determine that AKT1 but not AKT2 is required for the initiation of melanomagenesis using a BRAFV600E; ARF-/- GEM model.  Further, the authors show that inhibition of AKT2 expression leads to decreased expression of mRNAs encoding EMT-related proteins.  They further investigate the contribution of AKT2 to the regulation of glycolysis of melanoma cells such that knockdown of AKT2 impairs glycolysis and reduced levels of a critical regulator of glycolysis, pyruvate dehydrogenase kinase isoform 1 (PDHK1).  Overall, the experiments have been designed, performed and interpreted well but there are some important issues that the authors should address prior to publication:

Major points: 

1.     Line 233-234: The authors have not provided information about the number of shRNAs used to target each AKT isoform, but the guess is that only one was used.  The current standard for such experiments would normally be at least two independent shRNAs or evidence that an shRNA resistant form of the relevant gene can mitigate the effects of the shRNA.  Whereas we recognize that it would be unrealistic for the authors to repeat such rescue experiments hence the authors should acknowledge in the manuscript this possible technical weakness of their study in that they did not use multiple shRNAs against the various AKTs. 

2.     Line 225-226: The authors have presented quantification of immunoblots for pAKT isoforms in a panel of human melanoma cell lines without the blots itself.  We would encourage them to provide the immunoblots for at least the cell lines that were selected for the study.  We would also like them to present the PTEN immunoblots for these cell lines as this would complement the correlation study presented in Fig.S1C.

3.     Wound healing assay, which was used by the authors in one or more figures has not been described in the Methods section.

4.     Figure 1: The authors have depleted AKT isoforms in wound healing, cell migration and invasion assays.  It would be informative to know if overexpression of constitutively active AKT2 in cell lines that have no or low pAKT2 result in increased cell migration and invasion. 

5.     Fig 2D-E: The purpose of the in vivo experiment where the authors inject WM1799 shAKT2 cells subcutaneously into immunodeficient mice to assess anchorage independent growth has not been reasoned well.  Moreover, the data in Fig. 2E with AKT2 depletion and the data in Fig. S6E with AKT1 depletion through shRNA-mediated knockdown show decreased tumor growth.  Does this mean that both AKT1 and 2 are required for tumorigenicity?

6.     Fig. 2E and Fig. S4B are similar experiments with AKT isoform depleted either through shRNA or CRISPR/Cas9, respectively.  However, we see in Fig. 2E that AKT2 depletion using shRNA reduced tumor growth that was not recapitulated with CRISPR/Cas9-mediated AKT2 depletion in Fog. S4B. How do the authors explain these observations?

7.     290-294: The derivation and use of SM1-750 cell lines to increase the tumor ‘take rate’ is mentioned by the authors but were there any measures such as sequencing to ensure that these cells do not have additional mutations in other genes?

8.     Provide immunoblots of pAKT1 and pAKT3 for Figs. 2F, 3B and 3I as the activation status of these isoforms is unclear.

9.     Figure S3A: The immunoblots for the metastatic nodules shown in this figure do not have wells that represent SM1-750 cell lysates alone. Including this as a control will make the point stronger that the pAKT2 levels that are low or absent in SM1-750 cells have increased upon metastasis. In Rag2-/- mice. 

10.  Fig. 3: Since SM1-750 cells have undetectable pS474-AKT2 (Fig 3F), does overexpression of constitutively active AKT2 and subsequent tail vein injection lead to enhanced metastasis.  

11.  The authors should provide immunoblots for the cells with AKT isoform knockout achieved through CRISPR/Cas9. 

12.  365-367 and fig 5G: How is the prophylactic deletion of AKT2 using CRISPR/Cas9 not as effective as the knockdown experiments with respect to the survival study? There is some explanation provided in the discussion section but this has not been specifically addressed.

13.  Does reintroduction of AKT2 in the AKT2 KO cells rescue the phenotype of wound healing, cell migration, cell invasion and metastasis?

14.  The Kaplan-Meier survival curves for the GEM model of melanoma (Fig. S4D) arte not well explained.  How many mice were used in these experiments.  Moreover, since this model is based on BRAF(V600E)/ARF(Null) where does the driving force for PI3K activation come from that would then confer a dependency on AKT1?  Indeed, is there any evidence from analysis of tumor cell lysates that BRAF(V600E)/ARF(Null) melanomas have activation of PI3K>AKT signaling? 

15.  Does overexpression of AKT2 in a cell line with low AKT2 activity, increase EMT and invasion associated transcripts in the RT-qPCR analysis?

16.  Provide more details about how changes in glycolysis and EMT are related. These data seem rather tangential to the main thrust of the manuscript.  Moreover, since melanocytes are most certainly not of epithelial origin, why exactly does EMT mean in this context.  

Minor points:

1.          To maintain consistency between the figure panels, use the same labels.  For example, in Fig. S1D, the legend is not represented as shAKT 1/2/3, but figure panels in Fig.1 do have them represented as this. 

2.          Fig. S2C is missing '%’ is missing in the Y axis and Fig.S2D has not been mentioned in the legend for UACC903 cells. 

3.          Scale bar is missing in Fig. 1G

4.          Line 273-274 and Fig 3B: Provide the time point at which the immunoblotting was done to assess phospho-AKT2 in the metastatic lesions? Was it 6 weeks?  

5.          Fig 3D: It appears that 3 out of 4 mice in the image do not show any tumors in the lungs. Does this mean that they have no tumors or the imaging method used is unable to capture it? 

6.          Were lateral tail vein injections performed in all experiments that say tail vein injections? 

7.          287-288 and figure S3B: Does the SM1 cell line express ARF or is it silenced? SM1 cell line in the text is stated to be Arf-/-.  However, supplementary figure says that SM1 cell line has wild type Arf. 

8.          Fig. 3 legend is incomplete. 

9.          Fig. 5E-F: The luciferase images of the mice with AKT2 knockout provided in this figure panel do not show any metastatic tumors at 8 weeks. However, the quantification of this do show some metastatic nodules. How do the authors explain this?

10.       Fig. S4B and S4C: Was statistical analysis performed to determine significance? The number of mice used for these experiments has not been mentioned. 

11.       Can Ki67 staining be done on the tumor sections to assess the effect of AKT1 depletion on tumor growth?

12.       There are several mistakes throughout the discussion section in referencing the figures. 

Adequate

Author Response

I

In the manuscript by McRee et al., entitled “AKT2 loss impairs BRAF-mutant melanoma metastasis”, the authors have investigated the role of individual AKT isoforms in melanomagenesis and metastatic dissemination in BRAFV600E-driven melanoma cell lines and mouse models. Using a panel of BRAFV600E-driven human or mouse melanoma cell lines, they claim that knockdown of AKT2 inhibits cell migration and invasion. They performed tail vein injections of cells with reduced or ablated of AKT2 to assess the metastatic potential of AKT2 and claim that AKT2 depletion impairs metastasis. However, they determine that AKT1 but not AKT2 is required for the initiation of melanomagenesis using a BRAFV600E; ARF-/- GEM model. Further, the authors show that inhibition of AKT2 expression leads to decreased expression of mRNAs encoding EMT-related proteins. They further investigate the contribution of AKT2 to the regulation of glycolysis of melanoma cells such that knockdown of AKT2 impairs glycolysis and reduced levels of a critical regulator of glycolysis, pyruvate dehydrogenase kinase isoform 1
(PDHK1). Overall, the experiments have been designed, performed and interpreted well but there are some important issues that the authors should address prior to publication: Major points:

  1. Line 233-234: The authors have not provided information about the number of shRNAs used to target each AKT isoform, but the guess is that only one was used. The current standard for such experiments would normally be at least two independent shRNAs or evidence that an shRNA resistant form of the relevant gene can mitigate the effects of the shRNA. Whereas we recognize that it would be unrealistic for the authors to repeat such rescue experiments hence the authors should acknowledge in the manuscript this possible technical weakness of their study in that they did not use multiple shRNAs against the various AKTs.

Response: We agree with the reviewer that only using one shRNA is a limitation of our study, and now acknowledge this in the discussion in lines 537-540 of the revised manuscript. Importantly, we note that the hairpins chosen to produce the cell lines used in these studies were based on a well-validated series of such reagents produced and characterized by Alex Toker’s lab, and kindly provided to us by Dr. Toker (reference 34 in our revised manuscript).In addition, we feel that the similar phenotypes in wound healing, migration, invasion, and metastasis we show in both shRNA mediated depletion of AKT2 and CRISPR KO of AKT2 support that our conclusions on melanoma metastatic capability is dependent on specifically AKT knockout/knockdown.

  1. Line 225-226: The authors have presented quantification of immunoblots for pAKT isoforms in a panel of human melanoma cell lines without the blots itself. We would encourage them to provide the immunoblots for at least the cell lines that were selected for the study. We would also like them to present the PTEN immunoblots for these cell lines as this would complement the correlation study presented in Fig.S1C.

Response: Thank you for acknowledging this. To clarify and simplify these data, we now show a representative blot across human melanoma cell lines for PTEN levels, total AKT phosphorylation, and AKT2 phosphorylation, the focus of our study, in Figure S1A, as well as the correlation between PTEN and pAKT signal in Figure S1B. The phosphorylation of all AKT isoforms is not necessary for this study, and as such we decided to remove these data from the manuscript. The text in our revised manuscript now reflects this (lines 233-238).

  1. Wound healing assay, which was used by the authors in one or more figures has not been described in the Methods section.

Response: Thank you for pointing out this. We now include a section in the methods for the wound healing assay.

  1. Figure 1: The authors have depleted AKT isoforms in wound healing, cell migration and invasion assays. It would be informative to know if overexpression of constitutively active AKT2 in cell lines that have no or low pAKT2 result in increased cell migration and invasion.

Response: We agree in principle that ectopic activation of AKT2 might lead to increased cell migration and invasion in these cell lines, however, in practice we have found it very problematic to interpret forced expression of activated AKTs in our melanoma cells. Expression of constitutively active subunits may be lethal and can degrade what would otherwise appear to be isoform specific effects based on necessity assays. Therefore, we have been unable to assess the effects of overexpressed AKT2 on aspects of metastatic capability in this study. We acknowledge this in the discussion now on lines 577-580 of the discussion in the revised manuscript, and future studies that are beyond the scope of the current study will use precise genetic methods to optimize this system to more precisely understand the relationship between AKT2 activity and melanoma metastasis.

  1. Fig 2D-E: The purpose of the in vivo experiment where the authors inject WM1799 shAKT2 cells subcutaneously into immunodeficient mice to assess anchorage independent growth has not been reasoned well. Moreover, the data in Fig. 2E with AKT2 depletion and the data in Fig. S6E with AKT1 depletion through shRNA-mediated knockdown show decreased tumor growth. Does this mean that both AKT1 and 2 are required for tumorigenicity?

Response: The goal of assessing anchorage independent growth is to assess the seeding of melanoma cells in metastatic sites, as stated in lines 259-260 of the revised manuscript. Our data in 2E and S6E do support that both AKT1 and AKT2 are required for this aspect of tumorigenicity / metastatic seeding, which is discussed in lines 421-423 of the revised manuscript and lines 488-490 of the discussion, and shown in the graphical abstract of our manuscript.

  1. Fig. 2E and Fig. S4B are similar experiments with AKT isoform depleted either through shRNA or CRISPR/Cas9, respectively. However, we see in Fig. 2E that AKT2 depletion using shRNA reduced tumor growth that was not recapitulated with CRISPR/Cas9- mediated AKT2 depletion in Fog. S4B. How do the authors explain these observations?

in Fog. S4B. How do the authors explain these observations?

Response: Initially, we were surprised to see that while AKT2 knockout using CRISPR/Cas9 did recapitulate the reduced metastatic burden seen in the shRNA AKT2 KD cell line, the CRISPR mediated AKT2 knockout cells did not recapitulate the reduced tumor growth seen in the shRNA AKT2 KD cell line. We hypothesize that this might be a result of compensatory AKT1 phosphorylation that occurs in the complete absence of AKT2, but that does not immediately occur following acute knockdown of AKT2, and we discuss this in lines 560-564 of the revised manuscript discussion, although we did not explicitly test this hypothesis. Further, CRISPR knockout requires clonal outgrowth of individual cells, which might induce selective pressure, although we believe that while further characterization of these cells will be interesting, it is beyond the scope of this study.

  1. 290-294: The derivation and use of SM1-750 cell lines to increase the tumor ‘take rate’ is mentioned by the authors but were there any measures such as sequencing to ensure that these cells do not have additional mutations in other genes?

Response: We did not sequence the SM1-750 cell line to check for additional mutations, however we show a drastic increase in total AKT and AKT1 phosphorylation, and a minor increase in AKT3 phosphorylation in these cells compared to the parent SM1 cell line (Figure 3F). While we did not rule out the presence of other mutations, we believe the thorough characterization and robust phenotype of AKT activation status supports this as the mechanism for increased take rate in the context of our study focused on AKT activity in metastasis. Further, regardless of genetic background, the observation of frequent AKT2 activation in metastatic lesion vs the primary tumor population supports the concept of an important role for AKT2 in melanoma metastasis.

  1. Provide immunoblots of pAKT1 and pAKT3 for Figs. 2F, 3B and 3I as the activation status of these isoforms is unclear.

Response: We acknowledge that we do not show the activation status of AKT1 and AKT3 in the metastatic nodules referenced by the reviewer in these figures, and as we no longer have these samples we are unable to provide new analyses. However, given the extreme specificity for AKT2 in our in vitro characterization, and our focus on only AKT2 knockdown in vivo, we believe the phosphorylation of AKT2 in these experiments is sufficient for us to draw conclusions about the activity of AKT2 in melanoma metastasis. Future studies will interrogate the activity of other isoforms in metastatic capability. 

  1. Figure S3A: The immunoblots for the metastatic nodules shown in this figure do not have wells that represent SM1-750 cell lysates alone. Including this as a control will make the point stronger that the pAKT2 levels that are low or absent in SM1-750 cells have increased upon metastasis. In Rag2-/- mice.

Response: We acknowledge that the blots in Figure S3A do not have the SM1-750 lysate alone, however as we show the comparison between SM1-750 lysate and metastatic nodules from wildtype mice in Figure 3I, as well as the complete lack of AKT2 phosphorylation of the SM1-750 cells in Figure 3F, S3D, and S3E, we believe we are able to conclude that the same lack of phosphorylation was present in the SM1-750 cells before inoculation into Rag2-/- mice.

  1. Fig. 3: Since SM1-750 cells have undetectable pS474-AKT2 (Fig 3F), does overexpression of constitutively active AKT2 and subsequent tail vein injection lead to enhanced metastasis.

Response: We agree that in theory overexpression of AKT2 in this cell line might lead to enhanced metastasis, but due to the limitations described in our response to point 4 and in the discussion of our revised manuscript, we are currently unable to provide this. However, many studies in the literature support the concept that overexpression or constitutive activation of each of the AKT isoforms can promote tumor aggressiveness in a wide variety of tumor types, and note that our study is focused on the role of unmanipulated AKT2 in tumors driven by the BRAF mutation.

  1. The authors should provide immunoblots for the cells with AKT isoform knockout achieved through CRISPR/Cas9.

Response: We now show specific AKT2 knockout in new Figure 5A, as well as all isoform specific knockouts in Figure S4A, relevant to the experiments performed in each respective figure.

  1. 365-367 and fig 5G: How is the prophylactic deletion of AKT2 using CRISPR/Cas9 not as effective as the knockdown experiments with respect to the survival study? There is some explanation provided in the discussion section but this has not been specifically addressed.

Response: We agree that the “prophylactic deletion” using the AKT2 CRISPR KO cells shown in Figure 5 is not as robust as the shRNA knockdown shown in Figure 4. As explained in our response to point 6, we hypothesize this is a result of compensatory AKT1 activation, now mentioned in the discussion of our revised manuscript, and as above, we speculate that the ‘rewiring’ needed to compensate for acute loss of AKT2 in knockdown experiments is sufficiently delayed that the effects of certain AKT2 functions are observed at short times after knockdown but are overridden at longer times, or during the process of deriving the CRISPR KO cells. The nature of this rewiring and the preclinical impacts of it will be subject of future studies.

  1. Does reintroduction of AKT2 in the AKT2 KO cells rescue the phenotype of wound healing, cell migration, cell invasion and metastasis?

Response: As explained in our response to point 4 and point 10, we are currently unable to reintroduce or overexpress AKT2 into these cells and confidently interpret isoform specific differences.

  1. The Kaplan-Meier survival curves for the GEM model of melanoma (Fig. S4D) arte not well explained. How many mice were used in these experiments. Moreover, since this model is based on BRAF(V600E)/ARF(Null) where does the driving force for PI3K activation come from that would then confer a dependency on AKT1? Indeed, is there any evidence from analysis of tumor cell lysates that BRAF(V600E)/ARF(Null) melanomas have activation of PI3K>AKT signaling?

Response: We now add panel S4E which shows the number of mice for each group, median survival, and tumor incidence for this experiment. While our study does not explicitly evaluate the driving force of PI3K activation in this model, the isoform specific deletion of AKT1 impacting long term survival implies PI3K-AKT activation. We only performed survival studies in these mice, so are unable to analyze lysates, and we acknowledge this as a limitation in the revised manuscript on lines 502-503 of our revised manuscript.

  1. Does overexpression of AKT2 in a cell line with low AKT2 activity, increase EMT and invasion associated transcripts in the RT-qPCR analysis?

Response: As explained in our response to point 4, 10, and 13, we are currently unable to reintroduce or overexpress AKT2 into these cells and interpret isoform specific differences.

  1. Provide more details about how changes in glycolysis and EMT are related. These data seem rather tangential to the main thrust of the manuscript. Moreover, since melanocytes are most certainly not of epithelial origin, why exactly does EMT mean in this context.

Response: The relationship between glycolysis and EMT has been well established, showing elevated glycolytic enzyme activity (consistent with the Warburg effect) induces EMT signatures (cited in new references 55-56 and mentioned in lines 607-608 of our revised manuscript), and supporting our phenotype that AKT2 knockdown results in both decreased glycolytic activity and EMT-relates genes. Further, reference 56 shows an EMT signature of neural crest derived cells, such as melanocytes. In line with this, there is also a well established characterization of EMT-related gene signatures as drivers of melanoma metastasis (references 51-52).

Minor points:
1. To maintain consistency between the figure panels, use the same labels. For example, in Fig. S1D, the legend is not represented as shAKT 1/2/3, but figure panels in Fig.1 do have them represented as this.

Response: We have updated the legend in new Figure S1C (previously S1D) to use the same shAKT nomenclature used in Figure 1 and throughout the manuscript.

  1. Fig. S2C is missing '%’ is missing in the Y axis and Fig.S2D has not been mentioned in the legend for UACC903 cells.

Response: We have corrected the missing % in the Y axis and the figure legend including S2D.

  1. Scale bar is missing in Fig. 1G

Response: The full field of view is shown for 20x magnification, as indicated in the figure legend.

  1. Line 273-274 and Fig 3B: Provide the time point at which the immunoblotting was done to assess phospho-AKT2 in the metastatic lesions? Was it 6 weeks?

Response: Metastatic lesions were collected at the time of death, as such the lesions represent time points beginning around roughly 6 weeks up to 14 weeks, the duration of the studies shown in Figure 3E, which we now indicate in the figure legend.

  1. Fig 3D: It appears that 3 out of 4 mice in the image do not show any tumors in the lungs. Does this mean that they have no tumors or the imaging method used is unable to capture it?

Response: As indicated in Figure 3D, this imaging as performed at 6 weeks post inoculation. At this time point, all mice fed regular chow had robust metastasis and significant morbidity, while at this time point only one mouse fed Doxycycline chow had detectable metastasis (shown in Figure 3C-3D), and no mice had died yet. As the mice eventually succumb to death significantly later (by 14 weeks), we believe there are micrometastases present which are not detectable by luminescence imaging.

  1. Were lateral tail vein injections performed in all experiments that say tail vein injections?

Response: Lateral tail vein injections were performed, now stated in the methods.

  1. 287-288 and figure S3B: Does the SM1 cell line express ARF or is it silenced? SM1 cell line in the text is stated to be Arf-/-. However, supplementary figure says that SM1 cell line has wild type Arf.

Response: The SM1 parent cell line is ARF WT. The text has been corrected to reflect this, as shown in supplementary Figure 3B.

  1. Fig. 3 legend is incomplete.

Response: The full legend for figure 3 is now included.

  1. Fig. 5E-F: The luciferase images of the mice with AKT2 knockout provided in this figure panel do not show any metastatic tumors at 8 weeks. However, the quantification of this do show some metastatic nodules. How do the authors explain this?

Response: The luciferase images shown in Figure 5F we believe are most representative of the difference in metastasis between the NT and AKT2 KO cells, however additional mice not shown in those images had greater signal (as seen in the quantification in Figure 5G). Further, very minor signal is not well represented in these images, as such signal not able to be visualized in these images does still not quantify as 0.

  1. Fig. S4B and S4C: Was statistical analysis performed to determine significance? The number of mice used for these experiments has not been mentioned.

Response: We now show the number of mice for each of these panels (now Figure 5C and 5E respectively), and all statistical analysis was performed using Mantel-Cox tests, as indicated in the methods section.

  1. Can Ki67 staining be done on the tumor sections to assess the effect of AKT1 depletion on tumor growth?

Response: While we agree that Ki67 to assess in vivo proliferation would help support the conclusion that AKT1 loss impairs melanoma cell proliferation, we do not have tumor sections to perform this experiment. As such, we are concluding this based only on cell counting, cell cycle analysis, and BrDU incorporation.

  1. There are several mistakes throughout the discussion section in referencing the figures.

Response: We have updated the discussion and manuscript to reference all correct figures.

Round 2

Reviewer 2 Report

All my questions have been satisfactorily addressed in discussion or experimentally. In particular, adding the data of PDHK1 immunoprecipitation from the isoform specific CRIPSR knockout WM1799 cells has strengthened the story. 

Reviewer 4 Report

The authors are to be commended for having done such a solid job in responding to the critique.